# Convolutional Spike-triggered Covariance Analysis for Neural Subunit Models

**Anqi Wu**[1]        **Il Memming Park**[2]        **Jonathan W. Pillow**[1]

[1] Princeton Neuroscience Institute, Princeton University
{anqiw, pillow}@princeton.edu
[2] Department of Neurobiology and Behavior, Stony Brook University
memming.park@stonybrook.edu

## Abstract

Subunit models provide a powerful yet parsimonious description of neural responses to complex stimuli. They are defined by a cascade of two linear-nonlinear (LN) stages, with the first stage defined by a linear convolution with one or more filters and common point nonlinearity, and the second by pooling weights and an output nonlinearity. Recent interest in such models has surged due to their biological plausibility and accuracy for characterizing early sensory responses. However, fitting poses a difficult computational challenge due to the expense of evaluating the log-likelihood and the ubiquity of local optima. Here we address this problem by providing a theoretical connection between spike-triggered covariance analysis and nonlinear subunit models. Specifically, we show that a "convolutional" decomposition of a spike-triggered average (STA) and covariance (STC) matrix provides an asymptotically efficient estimator for class of quadratic subunit models. We establish theoretical conditions for identifiability of the subunit and pooling weights, and show that our estimator performs well even in cases of model mismatch. Finally, we analyze neural data from macaque primary visual cortex and show that our moment-based estimator outperforms a highly regularized generalized quadratic model (GQM), and achieves nearly the same prediction performance as the full maximum-likelihood estimator, yet at substantially lower cost.

## 1 Introduction

A central problem in systems neuroscience is to build flexible and accurate models of the sensory encoding process. Neurons are often characterized as responding to a small number of features in the high-dimensional space of natural stimuli. This motivates the idea of using dimensionality reduction methods to identify the features that affect the neural response [1–9]. However, many neurons in the early visual pathway pool signals from a small population of upstream neurons, each of which integrates and nolinearly transforms the light from a small region of visual space. For such neurons, stimulus selectivity is often not accurately described with a small number of filters [10]. A more accurate description can be obtained by assuming that such neurons pool inputs from an earlier stage of shifted, identical nonlinear "subunits" [11–13].

Recent interest in subunit models has surged due to their biological plausibility and accuracy for characterizing early sensory responses. In the visual system, linear pooling of shifted rectified linear filters was first proposed to describe sensory processing in the cat retina [14, 15], and more recent work has proposed similar models for responses in other early sensory areas [16–18]. Moreover, recent research in machine learning and computer vision has focused on hierarchical stacks of such subunit models, often referred to as Convolutional Neural Networks (CNN) [19–21].

The subunit models we consider here describe neural responses in terms of an LN-LN cascade, that is, a cascade of two linear-nonlinear (LN) processing stages, each of which involves linear projection and a nonlinear transformation. The first LN stage is convolutional, meaning it is formed from one or

more banks of identical, spatially shifted subunit filters, with outputs transformed by a shared subunit nonlinearity. The second LN stage consists of a set of weights for linearly pooling the nonlinear subunits, an output nonlinearity for mapping the output into the neuron's response range, and finally, an noise source for capturing the stochasticity of neural responses (typically assumed to be Gaussian, Bernoulli or Poisson). Vinch *et al* proposed one variant of this type of subunit model, and showed that it could account parsimoniously for the multi-dimensional input-output properties revealed by spike-triggered analysis of V1 responses [12, 13].

However, fitting such models remains a challenging problem. Simple LN models with Gaussian or Poisson noise can be fit very efficiently with spike-triggered-moment based estimators [6–8], but there is no equivalent theory for LN-LN or subunit models. This paper aims to fill that gap. We show that a convolutional decomposition of the spike-triggered average (STA) and covariance (STC) provides an asymptotically efficient estimator for a Poisson subunit model under certain technical conditions: the stimulus is Gaussian, the subunit nonlinearity is well

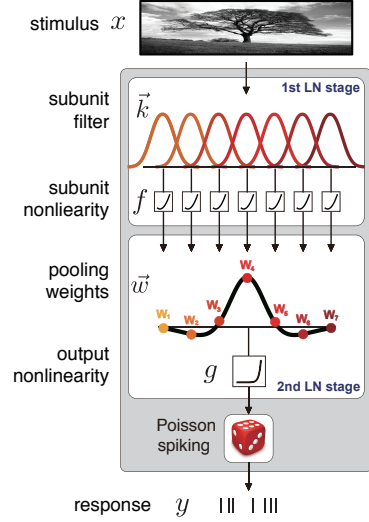

Figure 1: Schematic of subunit LN-LNP cascade model. For simplicity, we show only 1 subunit type.

described by a second-order polynomial, and the final nonlinearity is exponential. In this case, the subunit model represents a special case of a canonical Poisson generalized quadratic model (GQM), which allows us to apply the *expected log-likelihood* trick [7, 8] to reduce the log-likelihood to a form involving only the moments of the spike-triggered stimulus distribution. Estimating the subunit model from these moments, an approach we refer to as *convolutional STC*, has fixed computational cost that does not scale with the dataset size after a single pass through the data to compute sufficient statistics. We also establish theoretical conditions under which the model parameters are identifiable. Finally, we show that convolutional STC is robust to modest degrees of model mismatch, and is nearly as accurate as the full maximum likelihood estimator when applied to neural data from V1 simple and complex cells.

## 2 Subunit Model

We begin with a general definition of the Poisson convolutional subunit model (Fig. 1). The model is specified by:

$$\text{subunit outputs:} \quad s_{mi} = f(\mathbf{k}_m \cdot \mathbf{x}_i) \tag{1}$$

$$\text{spike rate:} \quad \lambda = g\Big(\sum_m \sum_i w_{mi}\, s_{mi}\Big) \tag{2}$$

$$\text{spike count:} \quad y|\lambda \sim \text{Poiss}(\lambda), \tag{3}$$

where $\mathbf{k}_m$ is the filter for the $m$'th type of subunit, $\mathbf{x}_i$ is the vectorized stimulus segment in the $i$'th position of the shifted filter during convolution, and $f$ is the nonlinearity governing subunit outputs. For the second stage, $w_{mi}$ is a linear pooling weight from the $m$'th subunit at position $i$, and $g$ is the neuron's output nonlinearity. Spike count $y$ is conditionally Poisson with rate $\lambda$.

Fitting subunit models with arbitrary $g$ and $f$ poses significant computational challenges. However, if we set $g$ to exponential and $f$ takes the form of second-order polynomial, the model reduces to

$$\lambda \;=\; \exp\Big(\tfrac{1}{2}\sum w_{mi}\,(\mathbf{k}_m \cdot \mathbf{x}_i)^2 + \sum w_{mi}\,(\mathbf{k}_m \cdot \mathbf{x}_i) + a\Big) \tag{4}$$

$$\;=\; \exp\Big(\quad \tfrac{1}{2}\,\mathbf{x}^\top C_{[\mathbf{w},\mathbf{k}]}\mathbf{x} \;+\; b_{[\mathbf{w},\mathbf{k}]}^\top \mathbf{x} \quad + a\Big) \tag{5}$$

where

$$C_{[\mathbf{w},\mathbf{k}]} = \sum_m K_m^\top \operatorname{diag}(\mathbf{w}_m) K_m, \qquad b_{[\mathbf{w},\mathbf{k}]} = \sum_m K_m^\top \mathbf{w}_m, \tag{6}$$

and $K_m$ is a Toeplitz matrix consisting of shifted copies of $\mathbf{k}_m$ satisfying $K_m \mathbf{x} = [\mathbf{x}_1, \mathbf{x}_2, \mathbf{x}_3, \ldots]^\top \mathbf{k}_m$.

In essence, these restrictions on the two nonlinearities reduce the subunit model to a (canonical-form) Poisson *generalized quadratic model* (GQM) [7, 8, 22], that is, a model in which the Poisson spike rate takes the form of an exponentiated quadratic function of the stimulus. We will pursue the implications of this mapping below. We assume that $\mathbf{k}$ is a spatial filter vector without time expansion. If we have a spatio-temporal stimulus-response, $\mathbf{k}$ should be a spatial-temporal filter, but the subunit convolution (across filter position $i$) involves only the spatial dimension(s). From (eqs. 4 and 5) it can be seen that the subunit model contains fewer parameters than a full GLM, making it a more parsimonious description for neurons with multi-dimensional stimulus selectivity.

## 3 Estimators for Subunit Model

With the above definitions and formulations, we now present three estimators for the model parameters $\{\mathbf{w}, \mathbf{k}\}$. To simplify the notation, we omit the subscript in $C_{[\mathbf{w},\mathbf{k}]}$ and $b_{[\mathbf{w},\mathbf{k}]}$, but their dependence on the model parameters is assumed throughout.

**Maximum Log-Likelihood Estimator**

The maximum log-likelihood estimator (MLE) has excellent asymptotic properties, though it comes with the high computational cost. The log-likelihood function can be written:

$$
\begin{aligned}
\mathcal{L}_{\text{MLE}}(\theta) &= \sum_i y_i \log \lambda_i - \sum_i \lambda_i && (7) \\
&= \sum y_i (\tfrac{1}{2}\mathbf{x}_i^\top C \mathbf{x}_i + b^\top \mathbf{x}_i + a) - \sum \exp(\tfrac{1}{2}\mathbf{x}_i^\top C \mathbf{x}_i + b^\top \mathbf{x}_i + a) && (8) \\
&= \text{Tr}[C\Lambda] + b^\top \mu + a n_{sp} - \left[\sum_i \exp(\tfrac{1}{2}\mathbf{x}_i^\top C \mathbf{x}_i + b^\top \mathbf{x}_i + a)\right] && (9)
\end{aligned}
$$

where $\mu = \sum_i y_i \mathbf{x}_i$ is the spike-triggered average (STA) and $\Lambda = \sum_i y_i \mathbf{x}_i \mathbf{x}_i^\top$ is the spike-triggered covariance (STC) and $n_{sp} = \sum_i y_i$ is the total number of spikes. We denote the MLE as $\theta_{\text{MLE}}$.

**Moment-based Estimator with Expected Log-Likelihood Fitting**

If the stimuli are drawn from $\mathbf{x} \sim \mathcal{N}(0, \Phi)$, a zero-mean Gaussian with covariance $\Phi$, then the expression in square brackets divided by $N$ in (eq. 9) will converge to its expectation, given by

$$
\mathbb{E}\left[\exp(\tfrac{1}{2}\mathbf{x}_i^\top C \mathbf{x}_i + b^\top \mathbf{x}_i + a)\right] = |I - \Phi C|^{-\frac{1}{2}} \exp\left(\tfrac{1}{2} b^\top (\Phi^{-1} - C)^{-1} b + a\right) \tag{10}
$$

Substituting this expectation into (9) yields a quantity called *expected log-likelihood*, with the objective function as,

$$
\mathcal{L}_{\text{ELL}}(\theta) = \text{Tr}[C\Lambda] + b^\top \mu + a n_{sp} - N|I - \Phi C|^{-\frac{1}{2}} \exp\left(\tfrac{1}{2} b^\top (\Phi^{-1} - C)^{-1} b + a\right) \tag{11}
$$

where $N$ is the number of time bins. We refer to $\theta_{\text{MELE}} = \arg\max_\theta \mathcal{L}_{\text{ELL}}(\theta)$ as the MELE (maximum expected log-likelihood estimator) [7, 8, 22].

**Moment-based Estimator with Least Squares Fitting**

Maximizing (11) w.r.t $\{C, b, a\}$ yields analytical expected maximum likelihood estimates [7]:

$$
C_{mele} = \Phi^{-1} - \Lambda^{-1}, \; b_{mele} = \Lambda^{-1}\mu, \; a_{mele} = \log(\tfrac{n_{sp}}{N}|\Phi\Lambda^{-1}|^{\frac{1}{2}}) - \tfrac{1}{2}\mu^\top \Phi^{-1}\Lambda^{-1}\mu \tag{12}
$$

With these analytical estimates, it is straightforward and to optimize $\mathbf{w}$ and $\mathbf{k}$ by directly minimizing squared error:

$$
\mathcal{L}_{LS}(\theta) = ||C_{mele} - K^\top \text{diag}(\mathbf{w})K||_2^2 + ||b_{mele} - K^\top \mathbf{w}||_2^2 \tag{13}
$$

which corresponds to an optimal "convolutional" decomposition of the moment-based estimates. This formulation shows that the eigenvectors of $C_{mele}$ are spanned by shifted copies of $\mathbf{k}$. We denote this estimate $\theta_{\text{LS}}$.

All three estimators, $\theta_{\text{MLE}}$, $\theta_{\text{MELE}}$ and $\theta_{\text{LS}}$ should provide consistent estimates for the subunit model parameters due to consistency of ML and MELE estimates. However, the moment-based estimates

(MELE and LS) are computationally much simpler, and scale much better to large datasets, due to the fact that they depend on the data only via the spike-triggered moments. In fact their only dependence on the dataset size is the cost of computing the STA and STC in one pass through the data. As for efficiency, $\theta_{\text{LS}}$ has the drawback of being sensitive to noise in the $C_{mele}$ estimate, which has far more free parameters than in the two vectors $\mathbf{w}$ and $\mathbf{k}$ (for a 1-subunit model). Therefore, accurate estimation of $C_{mele}$ should be a precondition for good performance of $\theta_{\text{LS}}$, and we expect $\theta_{\text{MELE}}$ to perform better for small datasets.

## 4 Identifiability

The equality $C = C_{[\mathbf{w},\mathbf{k}]} = K^\top \operatorname{diag}(\mathbf{w})K$ is a core assumption to bridge the theoretical connection between a subunit model and the spike-triggered moments (STA & STC). In case we care about recovering the underlying biological structure, we maybe interested to know when the solution is unique and naively interpretable. Here we address the identifiability of the *convolution decomposition* of $C$ for $\mathbf{k}$ and $\mathbf{w}$ estimation. Specifically, we briefly study the uniqueness of the form $C = K^\top \operatorname{diag}(\mathbf{w})K$ for a single subunit and multiple subunits respectively. We provide the proof for the single subunit case in the main text, and the proof for multiple subunits sharing the same pooling weight $\mathbf{w}$ in the supplement.

Note that failure of identifiability only indicates that there are possible symmetries in the solution space so that there are multiple equivalent optima, which is a question of theoretical interest, but it holds no implications for practical performance.

### 4.1 Identifiability for Single Subunit Model

We will frequently make use of frequency domain representation. Let $B \in \mathbb{R}^{d \times d}$ denote the discrete Fourier transform (DFT) matrix with $j$-th column is,

$$\mathbf{b}_j = \left[ 1, e^{-\frac{2\pi}{d}(j-1)}, e^{-\frac{2\pi}{d}2(j-1)}, e^{-\frac{2\pi}{d}3(j-1)}, \ldots, e^{-\frac{2\pi}{d}(d-1)(j-1)} \right]^\top. \tag{14}$$

Let $\widetilde{\mathbf{k}}$ be a $d$-dimensional vector resulting from a discrete Fourier transform, that is, $\widetilde{\mathbf{k}} = B_k\mathbf{k}$ where $B_k$ is a $d \times d_k$ DFT matrix, and similarly $\widetilde{\mathbf{w}} \in \mathbb{R}^d$ be a Fourier representation of $\mathbf{w}$.

We assume that $\mathbf{k}$ and $\mathbf{w}$ have full support in the frequency domain.

**Assumption 1.** *No element in $\widetilde{\mathbf{k}}$ or $\widetilde{\mathbf{w}}$ is zero.*

**Theorem.** *Suppose Assumption 1 holds, the convolution decomposition $C = K^\top \operatorname{diag}(\mathbf{w})K$ is uniquely identifiable up to shift and scale, where $C \in \mathbb{R}^{d \times d}$ and $d = d_k + d_w - 1$.*

*Proof.* We fix $\mathbf{k}$ (and thus $\widetilde{\mathbf{k}}$) to be a unit vector to deal with the obvious scale invariance. First note that we can rewrite the convolution operator $K$ using DFT matrices as,

$$K = B^H \operatorname{diag}(B_k\mathbf{k})B_w \tag{15}$$

where $B \in \mathbb{R}^{d \times d}$ is the DFT matrix and $(\cdot)^H$ denotes conjugate transpose operation. Thus,

$$C = B^H \operatorname{diag}(\widetilde{\mathbf{k}})^H \, B_w \operatorname{diag}(\mathbf{w})B_w^H \, \operatorname{diag}(\widetilde{\mathbf{k}})B \tag{16}$$

Note that $\widetilde{W} := B_w \operatorname{diag}(\mathbf{w})B_w^H$ is a circulant matrix,

$$\widetilde{W} := \operatorname{circulant}(\widetilde{\mathbf{w}}) = \begin{pmatrix} \widetilde{w}_1 & \widetilde{w}_d & \cdots & \widetilde{w}_3 & \widetilde{w}_2 \\ \widetilde{w}_2 & \widetilde{w}_1 & \cdots & \widetilde{w}_4 & \widetilde{w}_3 \\ \vdots & \vdots & \ddots & \vdots & \vdots \\ \widetilde{w}_{d-1} & \widetilde{w}_{d-2} & \cdots & \widetilde{w}_1 & \widetilde{w}_d \\ \widetilde{w}_d & \widetilde{w}_{d-1} & \cdots & \widetilde{w}_2 & \widetilde{w}_1 \end{pmatrix} \tag{17}$$

Hence, we can rewrite (16) in the frequency domain as,

$$\widetilde{C} = BCB^H = \operatorname{diag}(\widetilde{\mathbf{k}})^H \, \widetilde{W} \, \operatorname{diag}(\widetilde{\mathbf{k}}) = \widetilde{W} \odot (\widetilde{\mathbf{k}}\widetilde{\mathbf{k}}^H)^\top \tag{18}$$

Since $B$ is invertible, the uniqueness of the original $C$ decomposition is equivalent to the uniqueness of $\widetilde{C}$ decomposition. The newly defined decomposition is

$$\widetilde{C} = \widetilde{W} \odot (\widetilde{\mathbf{k}}\widetilde{\mathbf{k}}^H)^\top. \tag{19}$$

Suppose there are two distinct decompositions $\{\widetilde{W}, \widetilde{\mathbf{k}}\}$ and $\{\widetilde{V}, \widetilde{\mathbf{g}}\}$, where both $\{\mathbf{k}, \widetilde{\mathbf{k}}\}$ and $\{\mathbf{g}, \widetilde{\mathbf{g}}\}$ are unit vectors, such that $\widetilde{C} = \widetilde{W} \odot (\widetilde{\mathbf{k}}\widetilde{\mathbf{k}}^H)^\top = \widetilde{V} \odot (\widetilde{\mathbf{g}}\widetilde{\mathbf{g}}^H)^\top$. Since both $\widetilde{W}$ and $\widetilde{V}$ have no zero, define the element-wise ratio $R := (\widetilde{W}./\widetilde{V})^\top \in \mathbb{R}^{d \times d}$, then we have

$$R \odot \widetilde{\mathbf{k}}\widetilde{\mathbf{k}}^H = \widetilde{\mathbf{g}}\widetilde{\mathbf{g}}^H \tag{20}$$

Note that $\operatorname{rank}(R \odot \widetilde{\mathbf{k}}\widetilde{\mathbf{k}}^H) = \operatorname{rank}(\widetilde{\mathbf{g}}\widetilde{\mathbf{g}}^H) = 1$.

$R$ is also a circulant matrix which can be diagonalized by DFT [23]: $R = B \operatorname{diag}(r_1, \ldots, r_d) B^H$. We can express $R$ as $R = \sum_{i=1}^{d} r_i \mathbf{b}_i \mathbf{b}_i^H$. Using the identity for Hadamard product that for any vector $a$ and $b$, $(aa^H) \odot (bb^H) = (a \odot b)(a \odot b)^H$, we get

$$R \odot \widetilde{\mathbf{k}}\widetilde{\mathbf{k}}^H = \sum_{i=1}^{d} r_i(\mathbf{b}_i \mathbf{b}_i^H) \odot (\widetilde{\mathbf{k}}\widetilde{\mathbf{k}}^H) = \sum_{i=1}^{d} r_i(\mathbf{b}_i \odot \widetilde{\mathbf{k}})(\mathbf{b}_i \odot \widetilde{\mathbf{k}})^H \tag{21}$$

By Lemma 1 (in the appendix), $\{\mathbf{b}_1 \odot \widetilde{\mathbf{k}}, \mathbf{b}_2 \odot \widetilde{\mathbf{k}}, \ldots, \mathbf{b}_d \odot \widetilde{\mathbf{k}}\}$ is a linearly independent set.

Therefore, to satisfy the rank constraint $\operatorname{rank}(R \odot \widetilde{\mathbf{k}}\widetilde{\mathbf{k}}^H) = 1$, $r_i$ can be non-zero at most a single $i$. Without loss of generality, let $r_i \neq 0$ and all other $r.$ to be zero, then we have,

$$r_i(\mathbf{b}_i \mathbf{b}_i^H) \odot \widetilde{\mathbf{k}}\widetilde{\mathbf{k}}^H = \widetilde{\mathbf{g}}\widetilde{\mathbf{g}}^H \implies r_i \operatorname{diag}(\mathbf{b}_i)\widetilde{\mathbf{k}}\widetilde{\mathbf{k}}^H \operatorname{diag}(\mathbf{b}_i)^H = \widetilde{\mathbf{g}}\widetilde{\mathbf{g}}^H \tag{22}$$

Because $\mathbf{b}_i$, $\widetilde{\mathbf{k}}$ and $\widetilde{\mathbf{g}}$ are unit vectors, $r_i = 1$. By recognizing that $\left(\operatorname{diag}(\mathbf{b}_i)\widetilde{\mathbf{k}}\right)$ is the Fourier transform of $i-1$ positions shifted $\mathbf{k}$, denoted as $\mathbf{k}^{i-1}$, we have, $\mathbf{k}^{i-1}(\mathbf{k}^{i-1})^\top = \mathbf{g}\mathbf{g}^\top$. Therefore, $\mathbf{g} = \mathbf{k}^{i-1}$. Moreover, from (20) and (22), $(\mathbf{b}_i \mathbf{b}_i^H) \odot \widetilde{V} = \widetilde{W}$ thus, $\mathbf{v}^{i-1} = \mathbf{w}$. that is, $\mathbf{v}$ must also be a shifted version of $\mathbf{w}$.

If restricting $\mathbf{k}$ and $\mathbf{g}$ to be unit vectors, then any solution $\mathbf{v}$ and $\mathbf{g}$ would satisfy $\mathbf{w} = \mathbf{v}^{i-1}$ and $\mathbf{g} = \mathbf{k}^{i-1}$. Therefore, the two decompositions are identical up to scale and shift. $\quad\square$

## 4.2 Identifiability for Multiple Subunits Model

Multiple subunits model (with $m > 1$ subunits) is far more complicated to analyze due to large degree of hidden invariances. In this study, we only provide the analysis under a specific condition when all subunits share a common pooling weight $\mathbf{w}$.

**Assumption 2.** *All models share a common* $\mathbf{w}$.

We make a few additional assumptions. We would like to consider a tight parameterization where no combination of subunits can take over another subunit's task.

**Assumption 3.** $\mathbf{K} := [\mathbf{k}_1, \mathbf{k}_2, \mathbf{k}_3, \ldots, \mathbf{k}_m]$ *spans an $m$-dimensional subspace where $\mathbf{k}_i$ is the subunit filter for $i$-th subunit and* $\mathbf{K} \in \mathbb{R}^{d_k \times m}$. *In addition,* $\mathbf{K}$ *has orthogonal columns.*

We denote $\mathbf{K}$ with $p$ positions shifted along the column as $\mathbf{K}^p := [\mathbf{k}_1^p, \mathbf{k}_2^p, \mathbf{k}_3^p, \ldots, \mathbf{k}_m^p]$. Also, note that trivially, $m \leq d_k < d_k + d_w - 1 < d$ since $d_w > 1$.

To allow arbitrary scale corresponding to each unit vector $\mathbf{k}_i$, we introduce coefficient $\alpha_i$ to the $i$-th subunit, thus extending (19) to,

$$C = \sum_{i=1}^{m} \widetilde{W} \odot (\alpha_i \widetilde{\mathbf{k}}_i \widetilde{\mathbf{k}}_i^H)^\top = \widetilde{W} \odot \left(\sum_{i=1}^{m} \alpha_i \widetilde{\mathbf{k}}_i \widetilde{\mathbf{k}}_i^H\right)^\top = \widetilde{W} \odot (\widetilde{\mathbf{K}}\mathbf{A}\widetilde{\mathbf{K}}^H)^\top \tag{23}$$

where $\mathbf{A} \in \mathbb{R}^{m \times m}$ is a diagonal matrix of $\alpha_i$ and $\widetilde{\mathbf{K}} \in \mathbb{R}^{d \times m}$ is the DFT of $\mathbf{K}$.

**Assumption 4.** $\nexists \Omega \in \mathbb{R}^{m \times m}$ *such that* $\mathbf{K}^i \Omega = P\mathbf{K}^i$, $\forall i$, *where* $P \in \mathbb{R}^{d_k \times d_k}$ *is the permutation matrix from $\mathbf{K}^i$ to $\mathbf{K}^j$ by shifting rows, namely $\mathbf{K}^j = P\mathbf{K}^i$, $\forall i, j$, and $\Omega$ is a linear projection coefficient matrix satisfying* $\mathbf{K}^j = \mathbf{K}^i \Omega$.

**Assumption 5.** $\mathbf{A}$ *has all positive or all negative values on the diagonal.*

Given these assumptions, we establish the proposition for multiple subunits model.

**Proposition.** *Under Assumptions (1-5), the convolutional decomposition $C = \widetilde{W} \odot (\widetilde{\mathbf{K}}\mathbf{A}\widetilde{\mathbf{K}}^H)^\top$ is uniquely identifiable up to shift and scale.*

The proof for the proposition and illustrations of Assumption 4-5 are in the supplement.

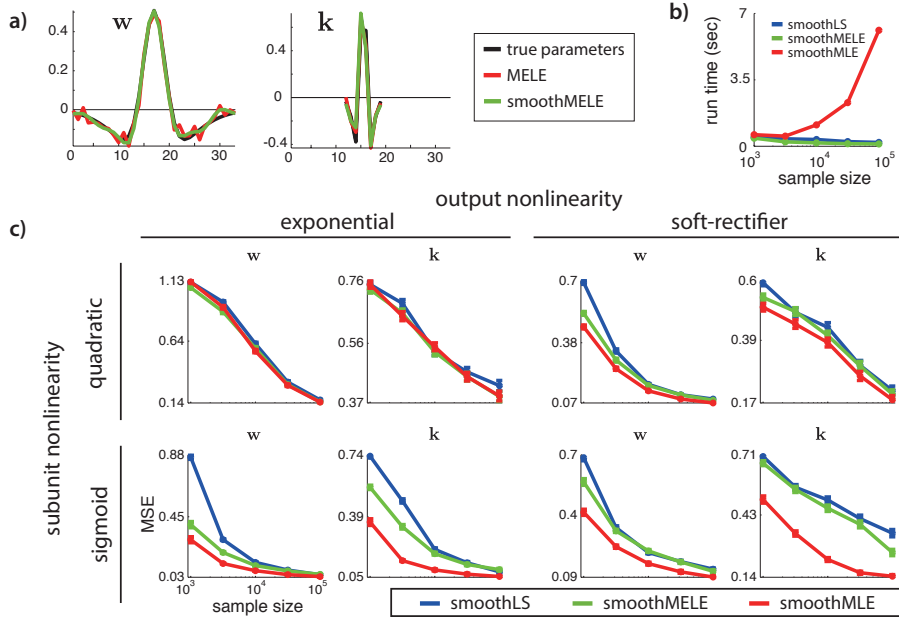

Figure 2: **a)** True parameters and MELE and smoothMELE estimations. **b)** Speed performance for smoothLS, smoothMELE and smoothMLE. The slightly decreasing running time along with a larger size is resulted from a more and more fully supported subspace, which makes optimization require fewer iterations. **c)** Accuracy performance for all combinations of subunit and output nonlinearities for smoothLS, smoothMELE and smoothMLE. Top left is the subunit model matching the data; others are model mismatch.

## 5 Experiments

### 5.1 Initialization

All three estimators are non-convex and contain many local optima, thus the selection of model initialization would affect the optimization substantially. Similar to [12] using 'convolutional STC' for initialization, we also use a simple moment based method with some assumptions. For simplicity, we assume all subunit models sharing the same $\mathbf{w}$ with different scaling factors as in eq. (23). Our initializer is generated from a shallow bilinear regression. Firstly, initialize $\mathbf{w}$ with a wide Gaussian profile, then estimate $\widetilde{\mathbf{K}}\mathbf{A}\widetilde{\mathbf{K}}^H$ from element-wise division of $C_{mele}$ by $\widetilde{W}$. Secondly, use SVD to decompose $\widetilde{\mathbf{K}}\mathbf{A}\widetilde{\mathbf{K}}^H$ into an orthogonal base set $\widetilde{\mathbf{K}}$ and a positive diagonal matrix $\mathbf{A}$, where $\widetilde{\mathbf{K}}$ and $\mathbf{A}$ contain information about $\mathbf{k}_i$'s and $\alpha$'s respectively, hypothesizing that $\mathbf{k}$'s are orthogonal to each other and $\alpha$'s are all positive (Assumptions 3 and 5). Based on the $\mathbf{k}_i$'s and $\alpha_i$'s we estimated from the rough Gaussian profile of $\mathbf{w}$, now we fix those and re-estimate $\mathbf{w}$ with the same element-wise division for $\widetilde{W}$. This bilinear iterative procedure proceeds only a few times in order to avoid overfitting to $C_{mele}$ which is a coarse estimate of $C$.

### 5.2 Smoothing prior

Neural receptive fields are generally smooth, thus a prior smoothing out high frequency fluctuations would improve the performance of estimators, unless the data likelihood provides sufficient evidence for jaggedness. We apply automatic smoothness determination (ASD [24]) to both $\mathbf{w}$ and $\mathbf{k}$, each with an associated balancing hyper parameter $\lambda_{\mathbf{w}}$ and $\lambda_{\mathbf{k}}$. Assuming $\mathbf{w} \sim \mathcal{N}(0, C_{\mathbf{w}})$ with

$$C_{\mathbf{w}} = \exp\left(-\rho_{\mathbf{w}} - \frac{\|\Delta\chi\|^2}{2\sigma_{\mathbf{w}}^2}\right) \tag{24}$$

where $\Delta\chi$ is the vector of differences between neighboring locations in $\mathbf{w}$. $\rho_{\mathbf{w}}$ and $\sigma_{\mathbf{w}}^2$ are variance and length scale of $C_{\mathbf{w}}$ that belong to the hyper parameter set. $\mathbf{k}$ also has the same ASD prior with hyper parameters $\rho_{\mathbf{k}}$ and $\sigma_{\mathbf{k}}^2$. For multiple subunits, each $\mathbf{w}_i$ and $\mathbf{k}_i$ would have its own ASD prior.

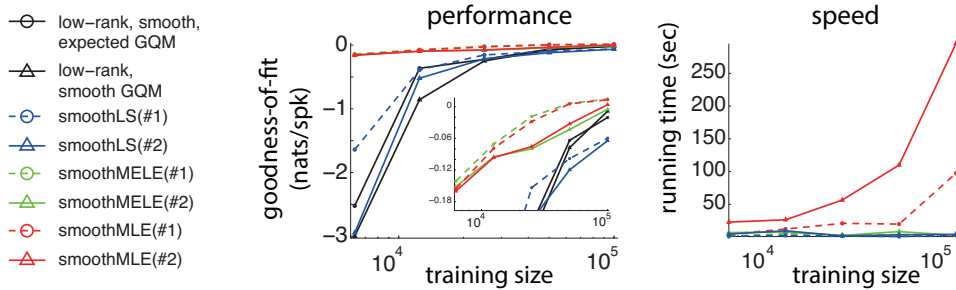

Figure 3: Goodness-of-model fits from various estimators and their running speeds (without GQM comparisons). Black curves are regularized GQM (with and without expected log-likelihood trick); blue is smooth LS; green is smooth MELE; red is smooth MLE. All the subunit estimators have results for 1 subunit and 2 subunits. The inset figure in performance is the enlarged view for large goodness-of-fit values. The right figure is the speed result showing that MLE-based methods require exponentially increasing running time when increasing the training size, but our moment-based ones have quite consistent speed.

Fig. 2a shows the true $\mathbf{w}$ and $\mathbf{k}$ and the estimations from MELE and smoothMELE (MELE with smoothing prior). From now on, we use smoothing prior by default.

### 5.3 Simulations

To illustrate the performance of our moment-based estimators, we generated Gaussian stimuli from an LNP neuron with exponentiated-quadratic nonlinearity and 1 subunit model with 8-element filter $\mathbf{k}$ and 33-element pooling weights $\mathbf{w}$. Mean firing rate is 0.91 spk/s. In our estimation, each time bin stimulus with 40 dimensions is treated as one sample to generate spike response. Fig. 2 b and c show the speed and accuracy performance of three estimators LS, MELE and MLE (with smoothing prior). LS and MELE are comparable with baseline MLE in terms of accuracy but are exponentially faster.

Although LNP with exponential nonlinearity has been widely adapted in neuroscience for its simplicity, the actual nonlinearity of neural systems is often sub-exponential, such as soft-rectifier nonlinearity. But exponential is favored as a convenient approximation of soft-rectifier within a small regime around the origin. Also generally, LNP neuron leans towards sigmoid subunit nonlinearity rather than quadratic. Quadratic could well approximate a sigmoid within a small nonlinear regime before the linear regime of the sigmoid. Therefore, in order to check the generalization performance of LS and MELE on mismatch models, we stimulated data from a neuron with sigmoid subunit nonlinearity or soft-rectifier output nonlinearity as shown in Fig. 2c. All the full MLEs formulated with no model mismatch provide a baseline for inspecting the performance of the ELL methods. Despite the model-mismatch, our estimators (LS and MELE) are on par with MLE when the subunit nonlinearity is quadratic, but the performance is notably worse for the sigmoid nonlinearity. Even so, in real applications, we will explore fits with different subunit nonlinearities using full MLE, where the exponential and quadratic assumption is thus primarily useful for a reasonable and extremely fast initializer. Moreover, the running time for moment-based estimators is always exponentially faster.

### 5.4 Application to neural data

In order to show the predictive performance more comprehensively in real neural dataset, we applied LS, MELE and MLE estimators to data from a population of 57 V1 simple and complex cells (data published in [11]). The stimulus consisted of oriented binary white noise ("flickering bar") aligned with the cell's preferred orientation. The size of receptive field was chosen to be # of bars $d \times 10$ time bins, yielding a $10d$-dimensional stimulus space. The time bin size is 10 ms and the number of bars ($d$) is 16 in our experiment.

We compared moment-based estimators and MLE with smoothed low-rank expected GQM and smoothed low-rank GQM [7, 8]. Models are trained on stimuli with size varying from $6.25 \times 10^3$ to $10^5$ and tested on $5 \times 10^4$ samples. Each subunit filter has a length of 5. All hyper parameters are chosen by cross validation. Fig. 3 shows that GQM is weakly better than LS but its running time is far more than LS (data not shown). Both MELE and MLE (but not LS) outfight GQM and

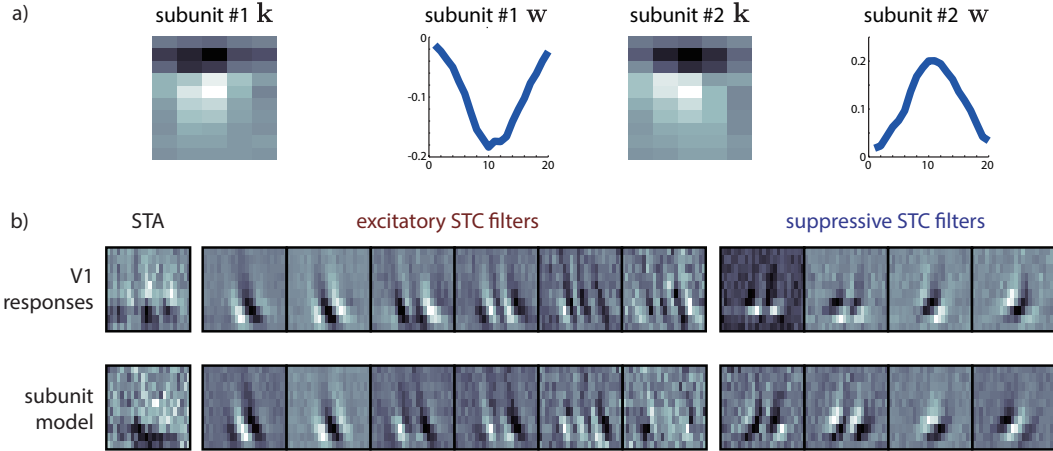

Figure 4: Estimating visual receptive fields from a complex cell (`5441029.p21`). **a**) $\mathbf{k}$ and $\mathbf{w}$ by fitting smoothMELE(#2). Subunit #1 is suppressive (negative $\mathbf{w}$) and #2 is excitatory (positive $\mathbf{w}$). Form the y-axis we can tell from $\mathbf{w}$ that both imply that middle subunits contribute more than the ends. **b**) Qualitative analysis. Each image corresponds to a normalized 24 dimensions spatial pixels (horizontal) by 10 time bins (vertical) filter. **Top row**: STA/STC from true data; **Bottom row**: simulated response from 2-subunit MELE model given true stimuli and applied the same subspace analysis.

expected GQM with both 1 subunit and 2 subunits. Especially the improvement is the greatest with 1 subunit, which results from the average over all simple and complex cells. Generally, the more "complex" the cell is, the higher probability that multiple subunits would fit better. Outstandingly, MELE outperforms others with best goodness-of-fit and flat speed curve. The goodness-of-fit is defined to be the log-likelihood on the test set divided by spike count.

For qualitative analysis, we ran smoothMELE(#2) for a complex cell and learned the optimal subunit filters and pooling weights (Fig. 4a), and then simulated V1 response by fitting 2-subunit MELE generative model given the optimal parameters. STA/STC analysis is applied to both neural data and simulated V1 response data. The quality of the filters trained on $10^5$ stimuli are qualitatively close to that obtained by STA/STC (Fig. 4b). Subunit models can recover STA, the first six excitatory STC filters and the last four suppressive ones but with a considerably parsimonious parameter space.

## 6 Conclusion

We proposed an asymptotically efficient estimator for quadratic convolutional subunit models, which forges an important theoretical link between spike-triggered covariance analysis and nonlinear subunit models. We have shown that the proposed method works well even when the assumptions about model specification (nonlinearity and input distribution) were violated. Our approach reduces the difficulty of fitting subunit models because computational cost does not depend on dataset size (beyond the cost of a single pass through the data to compute the spike-triggered moments). We also proved conditions for identifiability of the convolutional decomposition, which reveals that for most cases the parameters are indeed identifiable. We applied our estimators to the neural data from macaque primary visual cortex, and showed that they outperform a highly regularized form of the GQM and achieve similar performance to the subunit model MLE at substantially lower computational cost.

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
