[Supplementary Material · subunit_appendix.pdf]

# 7 Appendix

## 7.1 Identifiability for Single Subunit Model

**Lemma 1.** $[\mathbf{b}_1 \odot \widetilde{\mathbf{k}}, \mathbf{b}_2 \odot \widetilde{\mathbf{k}}, \dots, \mathbf{b}_d \odot \widetilde{\mathbf{k}}]$ *is a linearly independent set.*

If it is not linearly independent, there is a column vector $\mathbf{b}_p \odot \widetilde{\mathbf{k}}$ which is a linear combination of other vectors, thus

$$\mathbf{b}_p \odot \widetilde{\mathbf{k}} = \sum_{q \neq p} \alpha_q \mathbf{b}_q \odot \widetilde{\mathbf{k}} = (\sum_{q \neq p} \alpha_q \mathbf{b}_q) \odot \widetilde{\mathbf{k}} \tag{25}$$

Since $\widetilde{\mathbf{k}}$ has no zeros, we have

$$\mathbf{b}_p = \sum_{q \neq p} \alpha_q \mathbf{b}_q \tag{26}$$

where $\alpha_q$ is the arbitrary coefficient for vector $\mathbf{b}_q \odot \widetilde{\mathbf{k}}$. This contradicts that $B$ has orthogonal columns in (14). Thus $[\mathbf{b}_1 \odot \widetilde{\mathbf{k}}, \mathbf{b}_2 \odot \widetilde{\mathbf{k}}, \dots, \mathbf{b}_d \odot \widetilde{\mathbf{k}}]$ must be a linearly independent set and span a $d$-dimensional space.

## 7.2 Identifiability for Multiple Subunits Model with Same Pooling Weights

*Proof.* We also follow the similar contradiction proof as in single model situation by proving $\text{rank}(R) = 1$. Suppose there are multiple solutions,

$$C = \widetilde{W} \odot (\widetilde{\mathbf{K}} \mathbf{A} \widetilde{\mathbf{K}}^H)^\top = \widetilde{V} \odot (\widetilde{\mathbf{G}} M \widetilde{\mathbf{G}}^H)^\top \tag{27}$$

Since both $\widetilde{W}$ and $\widetilde{V}$ are assumed to have no zeros, let $R := (\widetilde{W}./\widetilde{V})^\top$, then we have

$$R \odot \widetilde{\mathbf{K}} \mathbf{A} \widetilde{\mathbf{K}}^H = \widetilde{\mathbf{G}} M \widetilde{\mathbf{G}}^H \tag{28}$$

Given that $R$ could be diagonalized by DFT and

$$\widetilde{\mathbf{K}} \mathbf{A} \widetilde{\mathbf{K}}^H = \sum_{i=1}^{m} \alpha_i \widetilde{\mathbf{k}}_i \widetilde{\mathbf{k}}_i^H, \quad \widetilde{\mathbf{G}} M \widetilde{\mathbf{G}}^H = \sum_{i=1}^{m} \beta_i \widetilde{\mathbf{g}}_i \widetilde{\mathbf{g}}_i^H \tag{29}$$

we can write

$$R \odot \widetilde{\mathbf{K}} \mathbf{A} \widetilde{\mathbf{K}}^H = \sum_{i=1}^{d} r_i \mathbf{b}_i \mathbf{b}_i^H \odot \sum_{i=1}^{m} \alpha_i \widetilde{\mathbf{k}}_i \widetilde{\mathbf{k}}_i^H \tag{30}$$

$$= \sum_{i=1}^{m} \sum_{j=1}^{d} r_j \alpha_i \mathbf{b}_j \mathbf{b}_j^H \odot \widetilde{\mathbf{k}}_i \widetilde{\mathbf{k}}_i^H \tag{31}$$

$$= \sum_{i=1}^{m} \sum_{j=1}^{d} r_j \alpha_i (\mathbf{b}_j \odot \widetilde{\mathbf{k}}_i)(\mathbf{b}_j \odot \widetilde{\mathbf{k}}_i)^H \tag{32}$$

Expanding $R \odot \widetilde{\mathbf{K}} \mathbf{A} \widetilde{\mathbf{K}}^H$ in a more explicit way, we have

$$R \odot \widetilde{\mathbf{K}} \mathbf{A} \widetilde{\mathbf{K}}^H = r_1 \alpha_1 (\mathbf{b}_1 \odot \widetilde{\mathbf{k}}_1)(\mathbf{b}_1 \odot \widetilde{\mathbf{k}}_1)^H + r_2 \alpha_1 (\mathbf{b}_2 \odot \widetilde{\mathbf{k}}_1)(\mathbf{b}_2 \odot \widetilde{\mathbf{k}}_1)^H + \dots + r_d \alpha_1 (\mathbf{b}_d \odot \widetilde{\mathbf{k}}_1)(\mathbf{b}_d \odot \widetilde{\mathbf{k}}_1)^H +$$
$$r_1 \alpha_2 (\mathbf{b}_1 \odot \widetilde{\mathbf{k}}_2)(\mathbf{b}_1 \odot \widetilde{\mathbf{k}}_2)^H + r_2 \alpha_2 (\mathbf{b}_2 \odot \widetilde{\mathbf{k}}_2)(\mathbf{b}_2 \odot \widetilde{\mathbf{k}}_2)^H + \dots + r_d \alpha_2 (\mathbf{b}_d \odot \widetilde{\mathbf{k}}_2)(\mathbf{b}_d \odot \widetilde{\mathbf{k}}_2)^H +$$
$$r_1 \alpha_3 (\mathbf{b}_1 \odot \widetilde{\mathbf{k}}_3)(\mathbf{b}_1 \odot \widetilde{\mathbf{k}}_3)^H + r_2 \alpha_3 (\mathbf{b}_2 \odot \widetilde{\mathbf{k}}_3)(\mathbf{b}_2 \odot \widetilde{\mathbf{k}}_3)^H + \dots + r_d \alpha_3 (\mathbf{b}_d \odot \widetilde{\mathbf{k}}_3)(\mathbf{b}_d \odot \widetilde{\mathbf{k}}_3)^H +$$
$$\vdots$$
$$r_1 \alpha_m (\mathbf{b}_1 \odot \widetilde{\mathbf{k}}_m)(\mathbf{b}_1 \odot \widetilde{\mathbf{k}}_m)^H + r_2 \alpha_m (\mathbf{b}_2 \odot \widetilde{\mathbf{k}}_m)(\mathbf{b}_2 \odot \widetilde{\mathbf{k}}_m)^H + \dots + r_d \alpha_m (\mathbf{b}_d \odot \widetilde{\mathbf{k}}_m)(\mathbf{b}_d \odot \widetilde{\mathbf{k}}_m)^H \tag{33}$$

Define $S_i := Span(\widetilde{\mathbf{K}}^{i-1}) = Span([\mathbf{b}_i \odot \widetilde{\mathbf{k}}_1, \mathbf{b}_i \odot \widetilde{\mathbf{k}}_2, \dots, \mathbf{b}_i \odot \widetilde{\mathbf{k}}_m])$ is a $m$-dimensional span for any $i$. $S_1 = Span(\widetilde{\mathbf{K}}) = Span([\mathbf{b}_1 \odot \widetilde{\mathbf{k}}_1, \mathbf{b}_1 \odot \widetilde{\mathbf{k}}_2, \dots, \mathbf{b}_1 \odot \widetilde{\mathbf{k}}_m])$.

If $\text{rank}(R) = 2$ with $r_i \neq 0$ and $r_j \neq 0$,

$$R \odot \widetilde{\mathbf{K}} \mathbf{A} \widetilde{\mathbf{K}}^H = r_i \alpha_1 (\mathbf{b}_i \odot \widetilde{\mathbf{k}}_1)(\mathbf{b}_i \odot \widetilde{\mathbf{k}}_1)^H + r_j \alpha_1 (\mathbf{b}_j \odot \widetilde{\mathbf{k}}_1)(\mathbf{b}_j \odot \widetilde{\mathbf{k}}_1)^H +$$

$$r_i\alpha_2(\mathbf{b}_i \odot \widetilde{\mathbf{k}}_2)(\mathbf{b}_i \odot \widetilde{\mathbf{k}}_2)^H + r_j\alpha_2(\mathbf{b}_j \odot \widetilde{\mathbf{k}}_2)(\mathbf{b}_j \odot \widetilde{\mathbf{k}}_2)^H +$$
$$r_i\alpha_3(\mathbf{b}_i \odot \widetilde{\mathbf{k}}_3)(\mathbf{b}_i \odot \widetilde{\mathbf{k}}_3)^H + r_j\alpha_3(\mathbf{b}_j \odot \widetilde{\mathbf{k}}_3)(\mathbf{b}_j \odot \widetilde{\mathbf{k}}_3)^H +$$
$$\vdots$$
$$r_i\alpha_m(\mathbf{b}_i \odot \widetilde{\mathbf{k}}_m)(\mathbf{b}_i \odot \widetilde{\mathbf{k}}_m)^H + r_j\alpha_m(\mathbf{b}_j \odot \widetilde{\mathbf{k}}_m)(\mathbf{b}_j \odot \widetilde{\mathbf{k}}_m)^H \tag{34}$$

To satisfy the rank of $R \odot \widetilde{\mathbf{K}}\mathbf{A}\widetilde{\mathbf{K}}^H$ to be $m$, we have

**Lemma 2.** $S_i = S_j$ when rank$(R) = 2$.

Since if $S_i \neq S_j$, there should be a vector $\mathbf{b}_j \odot \widetilde{\mathbf{k}}_p$ which cannot be represented as a linear combination of $[\mathbf{b}_i \odot \widetilde{\mathbf{k}}_1, \mathbf{b}_i \odot \widetilde{\mathbf{k}}_2, \ldots, \mathbf{b}_i \odot \widetilde{\mathbf{k}}_m]$ (same proof as Lemma 1), then rank$(R \odot \widetilde{\mathbf{K}}\mathbf{A}\widetilde{\mathbf{K}}^H) > m$. Thus $S_i$ and $S_j$ must be the same.

In addition, Lemma 2 implies that

**Corollary 1.** For any $p$, $S_p = S_{p+\delta}$, where $\delta = j - i$.

We now argue for multiple situations that given Corollary 1, rank$(R) = 1$ under the mild Assumption 4.

- If $\delta \nmid d$ ($\delta$ does not divide $d$), $\forall p$, $S_p = S_{p+\delta}$ means $S_1 = S_2 = \ldots = S_d$. All vectors $\forall j$, $\mathbf{b}_i \odot \widetilde{\mathbf{k}}_j$ lie in the same $m$-dimensional subspace. We also know that for any $i^{\text{th}}$ set, $[\mathbf{b}_1 \odot \widetilde{\mathbf{k}}_i, \mathbf{b}_2 \odot \widetilde{\mathbf{k}}_i, \ldots, \mathbf{b}_d \odot \widetilde{\mathbf{k}}_i]$ are linearly independent (Lemma 1) and span a $d$-dimensional space. Thus it induces a contradiction when $m < d$. A simpler illustration would be that it is impossible to claim that points in the same 2D space cannot spread out a 3D space, but the contrary holds. Therefore when $\delta \nmid d$, rank$(R) < 2 = 1$.

- If $\delta \mid d = \omega$, $S_p = S_{p+\delta}$ only indicates that $S_p = S_{p+\delta} = S_{p+2\delta} = \ldots = S_{p+d-\delta}$ ($\omega$ equal spans).

  - If $\omega > m$, this is similar to $\delta \nmid d$ case. That is, $[\mathbf{b}_p \odot \widetilde{\mathbf{k}}_i, \mathbf{b}_{p+\delta} \odot \widetilde{\mathbf{k}}_i, \mathbf{b}_{p+2\delta} \odot \widetilde{\mathbf{k}}_i, \ldots, \mathbf{b}_{p+d-\delta} \odot \widetilde{\mathbf{k}}_i]$ span an $\omega$-dimensional subspace which has higher dimension than $m$. But they also stay in the same $m$-dimensional subspace. Thus there is a contradiction and rank$(R) = 1$.

  - If $\omega \leq m$, it is possible that $R \odot \widetilde{\mathbf{K}}\mathbf{A}\widetilde{\mathbf{K}}^H$ consists of vectors from $\mathbf{K}^{i-1}$ and $\mathbf{K}^{j-1}$ with rank $m$. But $\mathbf{K}^{i-1}$ have the same column span with $\mathbf{K}^{j-1}$, because $S_i = S_j$. If $\mathbf{K}^{i-1}$ and $\mathbf{K}^{j-1}$ share the same column span, then there exists a linear projection coefficient matrix $\Omega$ satisfying $\mathbf{K}^{j-1} = \mathbf{K}^{i-1}\Omega$. Let $P$ be the permutation matrix from $\mathbf{K}^{i-1}$ to $\mathbf{K}^{j-1}$ by shifting rows, namely $\mathbf{K}^{j-1} = P\mathbf{K}^{i-1}$. This implies that we need to cook up a $\mathbf{K}$ whose projection matrix $\Omega$ for its $i-1$ shift $\mathbf{K}^{i-1}$ and $j-1$ shift $\mathbf{K}^{j-1}$ satisfies $\mathbf{K}^{i-1}\Omega = P\mathbf{K}^{i-1}$. In practice, this condition is barely satisfied. Thus, as long as $\nexists\Omega$, such that $\mathbf{K}^{i-1}\Omega = P\mathbf{K}^{i-1}$, $\mathbf{K}^{i-1}$ and $\mathbf{K}^{j-1}$ will not share the same span, then rank$(R \odot \widetilde{\mathbf{K}}\mathbf{A}\widetilde{\mathbf{K}}^H) > m$ conflicts with rank$(\widetilde{\mathbf{G}}M\widetilde{\mathbf{G}}^H) \leq m$. Consequently, rank$(R) = 1$. (This is the interpretation for Assumption 4.)

We can make similar arguments when rank$(R) > 2$, which only introduces more $m$-dimensional subspaces compared to rank$(R) = 2$ case. In sum, when rank$(R) \geq 2$, there is always a contradiction that rank$(R \odot \widetilde{\mathbf{K}}\mathbf{A}\widetilde{\mathbf{K}}^H) > $ rank$(\widetilde{\mathbf{G}}M\widetilde{\mathbf{G}}^H)$ if $\exists\Omega$ such that $\mathbf{K}^{i-1}\Omega = P\mathbf{K}^{i-1}$. Thus, there should be always rank$(R) = 1$.

Setting $r_i \neq 0$ and all others to be zero, we have

$$r_i(\mathbf{b}_i\mathbf{b}_i^H) \odot \widetilde{V} = \widetilde{W} \tag{35}$$

If we also assume both $\mathbf{w}$ and $\mathbf{v}$ are unit vectors to remove scaling vagueness, then $r_i = 1$, thus $\mathbf{w} = \mathbf{v}^{i-1}$.

We cannot claim the rigorous identifiability of $\mathbf{K}$ and $\mathbf{A}$, but we can claim

$$(\mathbf{b}_i \mathbf{b}_i^H) \odot \widetilde{\mathbf{K}} \mathbf{A} \widetilde{\mathbf{K}}^H = \widetilde{\mathbf{G}} M \widetilde{\mathbf{G}}^H \quad \Rightarrow \quad \operatorname{diag}(\mathbf{b}_i) \widetilde{\mathbf{K}} \mathbf{A} \widetilde{\mathbf{K}}^H \operatorname{diag}(\mathbf{b}_i)^H = \widetilde{\mathbf{G}} M \widetilde{\mathbf{G}}^H \tag{36}$$

$$\Rightarrow \quad \mathbf{K}^{i-1} \mathbf{A} (\mathbf{K}^{i-1})^\top = \mathbf{G} M \mathbf{G}^\top \tag{37}$$

When $\mathbf{A}$ has all positive or negative values and $\mathbf{K}$ has orthogonal columns (Assumption 3 and 5), the identifiability is reduced to the uniqueness of SVD, then $\mathbf{A}$ and $\mathbf{K}$ are both identifiable. $\quad\square$