[Reviews · NeurIPS 2015]

Submitted by Assigned_Reviewer_1

The authors present a moment-based method for estimating the nonlinear subunits of neurons in sensory systems. They begin by giving a formal definition of a convolutional neural subunit model and then proceed to show three methods for estimating the parameters of such a model. Finally, then present theorems for the identifyability of neural subunits, and show good performance in experiments with simulated and biological data.

The paper is clear and well written. It is a novel approach to estimation of neural subunit models. It provides a nice foundation for this class of work and is therefore significant to the field. There are no major concerns.

Minor questions: - Besides a decrease in MSE, what are the functional consequences for estimating the parameters of a model for data that is not generated with a quadratic subunit nonlinearity? - why does runtime go down with sample size in figure 2b? - can you clarify the following sentence on line 360? "All MLEs for each underlying model are equipped with the corresponding nonlinearity thus provides a baseline" - the binary white noise stimuli used in the biological experiment do not satisfy the Gaussian assumption. How does this affect the results? - can you explain the shape and sign of w in figure 4a?

Also, there is a typo in 'frequency' in line 196.
Summary: THe authors present a new model for framing and estimating the parameters of a neural subunit model. The work is novel and amounts to a significant advancement.

Submitted by Assigned_Reviewer_2

The authors link two types of models previously published at NIPS, i.e. canonical generalized quadratic model and subunit models, and thereby provide a novel parameter estimation technique for subunit models that is exponential faster compared to previous work (for similar accuracy).

This work is of relevance because it extends previous subunit models (like Vintch et. al) by introducing a likelihood approach based on a Poisson model. Surprisingly the new model is not compared to the previous model, but focuses on the computational efficiency of the GQM approach, although this was already pointed out extensively in previous work on "expected log-likelihood" and GQM.

The assumption of Gaussian input somewhat limits the approach to sensory input that can be controlled, in contrast to input from upstream cortical areas.

It should be pointed out why "identifiability" is important. Is it?

Quality and clarity are fine, although showing indices i and k in Fig. 1 and the removal of some typos would be beneficial for the reader: - Page 1, line 046, "Therefore, a..." says the opposite as intended - a not is missing. - Page 4, line 168, remove "and.
Summary: This work provides sufficient novelty in terms of algorithmic efficiency to justify a publication at NIPS.

Submitted by Assigned_Reviewer_3

The review has been modified based on the authors' feedback. Indeed, I have missed that Figure 3 showed performance on neural, and not simulated model.

This manuscript presents an approach for fitting linear-nonlinear models with spatial shifted linear filters to neural responses elicited by Gaussian stimuli. The overall idea is not new and is implemented here with very restrictive assumptions, such as that no fourier components of the pooling weight matrix are zero and specific type of nonlinearity is assumed. When tested on model cells generated according to the more typical sigmoidal nonlinearity, the algorithm perform worse than maximum likelihood estimators. The original publication of the dataset used by the authors quantitatively tested the spatial shifted subunit model.

It is worth noting that even if the assumption 1 on line 205 is satisfied, meaning that no fourier components of pooling filters are zero, the fact that the proof relies on the ratio between components suggests that the alogirthm will not be stable in cases where fourier components are small. I respectfully disagree with the authors' feedback that this issue does not have practical affects -- correct identification of subunits will affect how results are interpreted.
Summary: This manuscript presents an approach for estimating linear-nonlinear models with spatial shifted linear filters. The overall idea is not new and is implemented here with very restrictive assumptions, such as that no fourier components can be zero and specific type of nonlinearity is assumed. The proposed method performs worth than maximum likelihood even for a modest degree of mismatch between the observed and assumed subunit nonlinearity.

Submitted by Assigned_Reviewer_4

Some details:

figure 3, not sure what the goodness-of-fit is measured. (what does nats/spk mean?)

typo: line 46 does describe -> does not describe
Summary: Good paper that proposes fast algorithms for subunit model parameter estimation by approximating the likelihood function. Identifiability is discussed and experiment result is given to illustrate the strength of the fast estimator.

Author Feedback
Author rebuttal: We thank the reviewers for their careful reading of our paper and very constructive comments, which we feel will make the paper much stronger.

Our main concern is the score from reviewer #3. Based on this review, we realize we did not explain carefully enough how the main contributions fit together, and we apologize for this oversight.

As we see it, the paper has 3 primary contributions:
(1) a new connection between subunit models and GQM
(2) two new, fast moment-based fitting methods.
(3) A proof establishing conditions under which the model parameters are identifiable.

Reviewer 3's concerns focus on the restrictions in the proof for item #3. But these restrictions apply only to the identifiability question - they in no way undermine the practical utility of the model when identifiability fails to hold. Failure of identifiability just means that there are possible symmetries in the solution space so that there are multiple equivalent optima. This is a question of theoretical interest, but it holds no implications for practical performance. We will explain this point clearly in the revision, but in light of this misunderstanding we would like to (respectfully) ask the reviewer to reconsider their score.

==Reviewer 1==
- what are functional consequences for data not generated with a quadratic subunit nonlinearity?

In real datasets, we would like to explore fits with different subunit nonlinearities using full ML: the quadratic assumption is thus primarily a useful for reasonable (and extremely fast) initializer.

- runtime go down with sample size in figure 2b?

Great question - larger sample size gives a fully supported subspace, which makes optimization require fewer iterations. (It's a small effect, but we will add this point).

- sentence on line 360? ("provides a baseline")

Sorry, we meant to say that the full MLE here (formulated with no model mismatch) provides a baseline for inspecting the performance of the ELL methods (which will generally be worse, especially under mismatch, as in this example).

- binary white noise stimuli non-Gaussian

Thanks for noticing this - the fact that we do well in this setting shows that the method is also robust to stimulus assumptions.

- shape and sign of w in figure 4a

Subunit #1 is suppressive (negative w) and #2 is excitatory (positive w). Both imply that middle subunits contribute more than the ends. Thanks.

==Reviewer 2==
- Comparison with previous subunit models

Thanks for pointing this out. We didn't make these comparisons as the authors of the previous papers have not released code, but we would certainly like to carry these out in future work.

- Gaussian assumption limited to controlled input, not suitable for upstream cortical input

As demonstrated by the model mismatch experiment and RGC data, the Gaussian assumption is not critical. Moreover, ELL methods can provide initializer for ML estimation, speeding up convergence dramatically.

- should be pointed out why "identifiability" is important

We apologize for this oversight - we feel identifiability is of theoretical interest ("when is the solution unique"?) and in some cases we may care about the recovery of underlying biological structure. But it is not critical for prediction performance. (See discussion above).

==Reviewer 3==
- The overall idea is not new

We respectfully disagree. No one has connected subunit models with quadratic models, nor suggested moment-based estimators for them.

- Assumption of non-zero Fourier component

This only affects identifiability, not accuracy of the estimator or validity of the model.

- Instability caused by the ratio

Once again, this is not relevant to practical applications, only to identifiability. Although regularizing the estimates when the stimuli are rank deficient is an important practical element, standard regularization methods will work fine here.

- Specific type of nonlinearity is assumed

Please look again: we have explicitly considered model mismatch (where the nonlinearity differs from that assumed by the method) and also application to neural data.

- Performs worse than the MLE

Yes, this is expected - the MLE is the best-case here (i.e., since the fast methods are trying to approximate the MLE). The main point is speed (Fig 3B) -- we achieve a vast speedup for only a slight cost in accuracy.

- No quantitative comparison with experimental data is given

We apologize if we were not clear about emphasizing this, but please take another look: Fig 3 shows a quantitative comparison with 8 different methods using real neural data.

== Reviewer 4 ==

- goodness-of-fit
Sorry, this is log-likelihood on test set divided by spike count

== Reviewer 6 ==
Thanks, we will correct these errors.